# Extraction, Isolation, and Component Analysis of Turmeric-Derived Exosome-like Nanoparticles

**DOI:** 10.3390/bioengineering10101199

**Published:** 2023-10-15

**Authors:** Yongsheng Wei, Xiang Cai, Qiqi Wu, Hui Liao, Shuang Liang, Hongwei Fu, Qi Xiang, Shu Zhang

**Affiliations:** 1Guangdong Provincial Key Laboratory of Advanced Drug Delivery Systems, Center for Drug Research and Development of Guangdong Pharmaceutical University, Guangzhou 510006, China; yongsheng_gdpu@163.com (Y.W.); fuhon9we1@163.com (H.F.); 2Biopharmaceutical R&D Center of Jinan University Co., Ltd., Guangzhou 510632, China; 16626412540@163.com (X.C.); wuqiqi@stu2021.jnu.edu.cn (Q.W.); 18028636361@163.com (H.L.); 2021103898ls@stu2021.jnu.edu.cn (S.L.); 3Guangdong Provincial Key Laboratory of Bioengineering Medicine, Institute of Biomedicine, Jinan University, Guangzhou 510632, China

**Keywords:** turmeric-derived exosome-like nanoparticles (TELNs), multi-omics, component analysis

## Abstract

As one kind of plant-derived extracellular vesicle, turmeric-derived exosome-like nanoparticles (TELNs) are composed of proteins, lipids, nucleic acids, and small-molecule compounds, which possess good biocompatibility and safety. They are especially rich in information from the “mother plant”, which provides more applications in biological fields. In this study, we isolated and purified TELNs using differential centrifugation and ultracentrifugation and systematically detected their physicochemical properties using multi-omics. The TELNs possessed a typical teacup-like exosome morphology, and the extraction rate was approximately 1.71 ± 0.176 mg/g. The average particle size was 183.2 ± 10.9 nm, and the average zeta potential was −17.6 ± 1.19 mV. They were rich in lipids, mainly phosphatidylethanolamine (PE) (17.4%), triglyceride (TG) (12.3%), phosphatidylinositol (PI) (9.82%), and phosphatidylcholine (PC) (7.93%). All of them are the key lipids in the exosomes. The protein content was approximately 12% (M/M), mainly curcumin synthase and other proteins involved in secondary metabolite biosynthesis. In addition, there are critical essential genes for curcumin biosynthesis, such as curcumin synthase (CURS) and diketocoenzyme A synthase (DCS). More importantly, a greater variety of small-molecule compounds, primarily curcumin and curcumin analogs such as demethoxycurcumin and volatile oleoresins such as curcuminoids, have now been revealed. In conclusion, TELNs were successfully isolated, containing 0.17% (M/M) turmeric and a large amount of chemical information, the same as the parent-of-origin plant. This is the first time combining multi-omics to analyze the characteristics and nature of the TELNs, which laid a solid material foundation for the further development of turmeric.

## 1. Introduction

Turmeric (*Curcuma longa* L.), initially documented in China and featured in the Tang Materia Medica (659 A.D.), the world’s earliest pharmacopeia, is primarily found in the regions of Fujian, Taiwan, Guangdong, Guangxi, Sichuan, Yunnan, and Guizhou. Based on the 2020 edition of the Chinese Pharmacopoeia, turmeric is recognized for its therapeutic properties in promoting blood circulation and alleviating pain associated with menstruation. It is traditionally employed to address thoracic and hypochondriac pain, chest paralysis, dysmenorrhea, menstrual occlusion, abdominal obstruction, rheumatism, shoulder and arm pain, and swelling and pain resulting from injuries [1]. Given the reliance of the pharmacological characteristics of *Curcuma* spp. on their chemical compositions, investigations into the chemical constituents of turmeric/wild turmeric and other bioactive compounds have become increasingly important [2]. We can see that turmeric comprises curcuminoids, volatile oils, flavonoids, sugars, sterols, organic acids, and other constituents, exhibiting diverse biological activities [3]. The health benefits associated with turmeric primarily stem from the presence of a lipophilic polyphenol compound known as “curcumin”, which exhibits an orange-yellow hue and is derived from the rhizomes of the plant [4]. To achieve better curcumin, paying attention to the appropriate harvesting period for turmeric, typically from September to November, is crucial. During this time, the growth rate of turmeric rhizomes is slow, allowing for the gradual attainment of optimal levels of nutrients and other components. Moreover, given the historical cultivation of turmeric plants, turmeric in Guangxi is the best one [5]. In Guangxi province, to guarantee turmeric quality, standardized cultivation practices have been established. We are delighted to see that the turmeric industry is booming.

The exosome, composed of lipids, proteins, and nucleic acids, is a carrier with a double-membrane structure [6]. Medicinal plants have been integral to the culture of traditional Chinese medicine for millennia and serve as a foundation for the development of innovative drugs. Research indicates that rare nanoparticles can be extracted from plant juices, such as those of lemon [7], grape [8], ginger [9], and tomato [10]. These nanosized vesicles, known as exosome-like nanoparticles (ELNs) [11], possess a phospholipid bilayer structure resembling mammalian exosomes and exhibit a teacup-shaped morphology. Research has demonstrated that plant-derived ELNs possess the capacity to serve as drug delivery systems. These ELNs can be engineered for modification and lipid reorganization, thereby enabling their integration into newly engineered nanocarriers. However, as drug delivery vehicles, ELNs need to be characterized to reveal their pharmacological activity and drug-loading properties. Gao, C. et al. [12] studied turmeric-derived exosome-like nanoparticles and found that they contained the active ingredients of turmeric, for example, curcumin. Therefore, they loaded turmeric-derived exosome-like nanoparticles with astragalus components to enhance absorption and promote their antitumor effects.

In order to better study turmeric and turmeric-derived exosomes, we not only extracted turmeric-derived exosome-like nanoparticles (TELNs) from turmeric but also used multi-omics to comprehensively analyze the components in TELNs, laying a solid material foundation for the further development of turmeric and TELNs.

## 2. Materials and Methods

### 2.1. Instruments and Materials

Instruments: High speed refrigerated centrifuge (Sigma 3K15, Osterode am Harz, Germany), nanoparticle-tracking analyzer (NanoSight NS 3000, Malvern Panalytical, West Midlands, UK), nanoparticle size/zeta potential distribution analyzer (Beckman Coulter, Brea, CA, USA), transmission electron microscope (TECNAI G2 Spirit TWIN, Thermo Fisher Scientific, Waltham, MA, USA), atomic force microscope (Bioscope Catalyst/Multimode, Bruker, Karlsruhe, Germany), LC40 High-Performance Liquid Chromatograph (Shimadzu, Kyoto, Japan), GC-MS (Trace 1300ISQ LT, Thermo Fisher Scientific, USA), High-Resolution LC-MS (SYNAPT-G2 Q-TOF, Waters, Milford, MA, USA), Ultra-micro UV Spectrophotometer (Kai-O Technology Development Co., Ltd., Beijing, China), PCR Instrument (Langji Scientific Instrument Co., Ltd., Hangzhou, China), Real-time Fluorescence PCR Instrument (BIO-RAD, Hercules, CA, USA), and Eyela FDU-1200 freeze dryer (Eyela, Tokyo, Japan).

Materials: curcumin (≥98%) (McLean Biochemical Technology Co., Ltd., Shanghai, China), fresh turmeric (Guangxi, China), BCA kit (Thermo Fisher Scientific, USA), paraformaldehyde (Sinopharm Chemical Reagent Co., Ltd., Shanghai, China), glutaraldehyde (Guangzhou Chemical Reagent Factory, Guangzhou, China), phosphotungstic acid (McLean Biochemical Technology Co., Ltd., Shanghai, China), and phosphotungstic acid (McLean Biochemical Technology Co., Ltd., Shanghai, China).

### 2.2. Identification of Turmeric

For this study, turmeric sourced from the Guangxi region was chosen as the material and subjected to morphological identification by observing the fresh turmeric rhizomes’ whole, transverse, and longitudinal sections.

### 2.3. Extraction and Separation of TELNs

Approximately 600 g of fresh, peeled turmeric was weighed, squeezed, and filtered through a 200-mesh screen to remove fiber impurities. The juice of the fresh, peeled turmeric was subjected to centrifugation at 1000× *g* (10 min), 3000× *g* (20 min), and 10,000× *g* (40 min) at 4 °C. The supernatant was collected and filtered with a 0.8 μm filter. The filtrate was centrifuged at 100,000× *g* for 2 h at 4 °C, and then the supernatant was discarded, the pellet was resuspended in an appropriate amount of PBS, centrifuged again at 100,000× *g* for 2 h, and then the gel pellet was collected, weighing approximately 1 g. The gel precipitate was added to PBS and resuspended using ultrasonication, and the resuspension solution was filtered through 0.8 μm, 0.45 μm, and 0.22 μm microporous filter membranes to obtain the TELNs. A gel precipitate of 1.71 ± 0.176 mg TELNs per 1 g of peeled turmeric was isolated (*n* = 4) and stored at 4 °C for use.

### 2.4. Physical Characterization of TELNs

#### 2.4.1. Particle Size and Zeta Potential

TELNs were weighed (25 mg), added to 1 mL of ultrapure water, resuspended using sonication (40 kHz), and filtered. The particle size and distribution of TELNs were determined using a nanoparticle tracking analyzer (NTA) after dilution with ultrapure water (1:2000).

Subsequently, 0.5 mL of TELNs obtained after the above operation was added to 4.5 mL of ultrapure water and vortexed. DLS and a Malvern nanoparticle size/zeta potential distribution analyzer were used to detect and record the particle size and zeta potential.

#### 2.4.2. Appearance and Morphology

Atomic force microscopy (AFM): The filtrate obtained in Section 2.4.1 was diluted by a factor of 10 and then dispensed in 10 μL droplets onto a mica sheet. The sample was left to air-dry overnight, after which the morphology of the TELNs was observed using AFM. Photographs were taken and recorded for further analysis. NanoScope Analysis software was used to analyze the data.

Transmission electron microscopy (TEM): A total of 500 μL of TELNs was fixed with an equal volume of 4% paraformaldehyde for 30 min, 20 μL was aspirated in a Petri dish, and a copper mesh was placed on the droplet and incubated for 20 min. After the liquid was blotted with filter paper, the copper mesh was washed twice on a PBS droplet. After this, the mesh was fixed in a 1% glutaraldehyde solution droplet for 5 min and then washed in a droplet of distilled water 5 times. Finally, the mesh was stained with 5% phosphotungstic acid for 3 min. The copper mesh was placed under the TEM to observe the morphological structure of TELNs, photographed, and recorded.

### 2.5. Lipid Determination

Sample preparation: A total of 87.99 mg of TELNs was weighed precisely and resuspended in 1 mL of ultrapure water, then 5 mL of total lipid extraction reagent (trichloromethane: methanol = 2:1, *v*/*v*) was added and mixed by vortex shaking. The mixture was centrifuged at 2000× *g* for 10 min for complete stratification, and the organic layer was collected and blown dry under nitrogen. The precipitate was resuspended by adding 200 μL of methanol.

#### 2.5.1. Thin-Layer Chromatography (TLC)

A silica gel thin-layer plate (5 cm × 10 cm) and 9 mm capillary spotting were used, and the unfolding agent was trichloromethane/methanol/acetic acid = 190:9:1, *v*/*v*/*v*. The color developer was a 10% CuSO_4_ solution (the solvent was an 8% phosphoric acid solution). The sample was placed in a 100 °C oven to carbonize for 10 min and was observed and photographed.

#### 2.5.2. LC-MS-Based Lipidomic Analysis

The detection was performed using a tandem Q Exactive high-resolution mass spectrometer (Thermo Fisher Scientific, USA). The chromatographic column used was a CSH C18 (1.7 μm, 2.1 × 100 mm, Waters, USA). In positive ionization mode, the mobile phases consisted of an aqueous solution containing 10 mM ammonia formate, 0.1% formic acid, and 60% acetonitrile (liquid A) and a solution containing 10 mM ammonia formate, 0.1% formic acid, 90% isopropanol, and 10% ACN (liquid B). Additionally, in negative ionization mode, an aqueous solution containing 10 mM ammonia formate and 60% ACN (liquid A) and a solution containing 10 mM ammonia formate, 90% isopropanol, and 10% acetonitrile were used.

### 2.6. Protein Determination

#### 2.6.1. Protein Quantification and Analysis

The BCA method was employed to quantify the protein content of TELNs. The proteins in TELN samples were separated using routine SDS-PAGE analysis, utilizing a 5% concentrate gel and a 10% separator gel. Subsequently, the separated proteins were stained with Coomassie Brilliant Blue and Silver Nitrate, respectively, and were observed and photographed for documentation purposes.

#### 2.6.2. LC-MS-Based Proteomic Analysis

Sample preparation: TELNs were rapidly frozen in liquid nitrogen. Samples were lyophilized in a vacuum overnight using a Eyela FDU-1200 freeze dryer (Eyela, Tokyo, Japan). The lyophilized TELN samples were resolubilized and desalted using mobile phase A, which consisted of 2% ACN and 0.1% FA. The analysis was performed using a tandem mass spectrometer Q-Exactive HF X (Thermo Fisher Scientific, San Jose, CA, USA) equipped with a C18 column. The column had a diameter of 75 μm and a particle size of 3 μm, with a length of 25 μm. The mobile phase consisted of liquid A (2% ACN, 0.1% FA) and liquid B (98% ACN, 0.1% FA). The flow rate was set at 300 mL/min. Gradient: 0–5 min, 5% liquid B (98% ACN, 0.1% FA); 5–45 min, 5~25% liquid B; 45–50 min, 25~35% liquid B; 50–52 min, 35~80% B solution; 52–54 min, 80% B solution; and 54–60 min, 80~5% B solution.

The peptides that had been separated were subsequently subjected to ionization through a nano-electrospray ionization (nano-ESI) source and introduced into a tandem mass spectrometer Q-Exactive HF X (manufactured by Thermo Fisher Scientific, San Jose, CA, USA) for the purpose of detection. The resulting raw data files were then subjected to processing using the Mascot v2.3.02 data analysis software.

### 2.7. Nucleic Acid Determination

#### 2.7.1. Sample Preparation and Identification

Approximately 50 milligrams (mg) of TELNs were utilized for the purpose of extracting total RNA. The TELNs RNA was subsequently subjected to agarose gel electrophoresis, employing a voltage of 120 volts (V) for a duration of 15 min. Following this, gel imaging was conducted, enabling the observation and photographic documentation of the positions of the RNA bands.

#### 2.7.2. Real-Time Fluorescence Quantitative PCR (RT-qPCR)

RNA was synthesized using reverse transcription, after which 2.5 μL of the cDNA template was taken for PCR identification. The primers required for the experiment were designed by searching the literature [13], commissioned for synthesis by Beijing Genomics Institute (BGI; Shenzhen, China), and are shown in Table 1. Finally, the gene expression of TELNs was analyzed using real-time fluorescence quantitative PCR (RT-qPCR).

### 2.8. Determination of Chemical Composition

#### 2.8.1. LC-MS-Based Chemical Composition Analysis

Sample preparation: Approximately 50 mg of TELNs was accurately weighed into a 1.5 mL EP tube, followed by the addition of 500 μL of methanol. The mixture was subjected to ultrasonic disruption at a frequency of 40 kHz for a duration of 60 min. Subsequently, the supernatant layer was separated through centrifugation, and this extraction process was repeated three times. Finally, the resulting solution was subjected to filtration. Chromatographic conditions: The chromatographic column employed in this study was the Waters X Bridge C18, with dimensions of 2.1 mm × 150 mm and a particle size of 5 μm. The mobile phase consisted of water and 0.1% formic acid (FA) as component A and acetonitrile (ACN) and 0.1% FA as component B. The gradient elution method was employed for the analysis, with the following liquid composition and time intervals: 0–1 min, 5% B; 1.1–10 min, 5~80% B; 10.1–12 min, 80~100% B; 12.1–13 min, 12.1~100% B; 12.1–13 min, 100% B; 13.1–14 min, 100~5% B; 14–15 min, 5% B; and 13.1–15 min, 40% B. The flow rate used was 0.3 mL/min, and the column temperature was maintained at 40 °C. Each injection consisted of 10 μL of the sample. Mass spectrometry was performed in electrospray ionization (ESI) positive ion mode, using an electrospray ion source. The atomization air pressure was set at 35 psi, with a nitrogen flow rate of 12 L/min and a temperature of 325 °C. The ionization voltage applied was 4 kV, and the fragmentation voltage was set at 250 V in action mode.

#### 2.8.2. GC-MS-Based Volatile Components Analysis

Approximately 50 mg of TELNs was weighed into a 1.5 mL EP tube, and 500 μL of trichloromethane was added and ultrasonically broken at a frequency of 40 kHz for 30 min. The supernatant layer was centrifuged, the extraction was repeated three times, and GC-MS detected the volatile oil compositions.

GC conditions: DB-1 flexible quartz capillary column (30 m × 0.25 mm × 0.25 μm); carrier gas, high-purity nitrogen; volume flow rate, 1.0 mL/min; split ratio, 80:1; inlet temperature, 280 °C; injection volume, 0.5 μL; and initial temperature, 70 °C, increased to 260 °C at an increased rate of 5 °C/min and maintained for 10 min. Mass spectrometry conditions: HP5973 mass spectrometry detector (MSD); ionization mode, EI (ion source); electron energy, 70 eV; mass scanning range, *m*/*z* 33–800; and full scanning mode, the NIST mass spectral library.

#### 2.8.3. HPLC-Based for Quantification of Curcumin Analysis

Sample preparation: 50.9 g of TELNs and 51.1 g of peeled turmeric rhizomes were weighed and extracted with trichloromethane solvent. Chromatographic conditions: column, GLscience-C18 column (4.6 mm × 250 mm, 5 μm); mobile phase, acetonitrile-4% glacial acetic acid solution (48:52); flow rate, 1.0 mL/min; column temperature, 25 °C; detection wavelength, 430 nm; and injection volume, 10 μL.

### 2.9. Statistical Analysis

Data are presented as the mean ± S.E.M. Data graphics and statistical analysis were performed using GraphPad Prism 8.0. Mascot v2.3.02 software was used to identify proteins against UniProt, NCBI, and Ensembl databases and annotate them with GO, KOG, and Pathway.

## 3. Results

### 3.1. Morphological Identification of Turmeric

The morphological examination of the studied turmeric product revealed it to be irregularly ovoid, cylindrical, or pike-shaped, often curved, some with short, forked branches, 2~5 cm long and 1~3 cm in diameter, with a dark yellow surface, coarse sugar crystals, wrinkled texture, apparent links, and rounded branching and fibrous root scars (Figure 1a). The texture was solid and not easy to break; the section was brownish-yellow to golden-yellow, horn-like, with a waxy luster; the endothelial ring was prominent; and the vascular bundles were scattered in dots (Figure 1a,b).

### 3.2. Physical Characterization of TELNs

#### 3.2.1. Particle Size and Zeta Potential

The TELNs were concentrated around 180 nm (Figure 2a), and the particle size distribution showed a single-peak normal distribution (Figure 2b). The particle size of the turmeric ELNs obtained by the combination of differential centrifugation and ultracentrifugation was mainly distributed in the range of 80.0–400 nm, which was consistent with the literature report [14]. After the TELN gel precipitate was diluted 2000 times with PBS, the particle concentration detected using the Malvern particle size analyzer was 4.51 × 10^8^ ± 5.85 × 10^7^ particles/mL, the average particle size was 163.1 nm, and the PDI was 0.212. The surface of the TELNs was negatively charged with an average ζ-potential of −17.6 ± 1.19 mV (Figure 2c).

#### 3.2.2. Appearance and Morphology

The TELNs were observed to possess an apparent spherical nanoparticle structure, relatively uniform dispersion, and uniform morphology (Figure 3a), and after statistical analysis, it was found that the average particle size of the particles was 176.2 ± 54.47 nm (Figure 3b), which was consistent with the average size of the TELNs detected using the NTA (180 nm).

The TELNs showed a typical teacup-like double-membrane structure (Figure 3c) of exosomes.

### 3.3. Lipid Determination

The TLC results showed that the TELNs contained eight components with high content and within the detection limit of thin-layer chromatography, consistent with those contained in the rhizomes of turmeric. The bands of curcumin were detected in TELNs at the equivalent displacement of the curcumin control (Figure 4a).

The results of the lipidomics identification showed that a total of 615 lipids were identified in the TELNs, of which the top five were phosphatidylethanolamine (PE) and triglyceride (TG), phosphatidylinositol (PI), phosphatidylcholine (PC) and digalactosylglycerol (DGDG), with contents of 17.4%, 12.3%, 9.82%, 7.93% and 7.81%, respectively. (Figure 4b,c).

### 3.4. Protein Determination

As shown in Table 2, the results of the quantitative BCA assay indicated that 0.125 ± 0.0162 mg of protein (*n* = 7) was measured in each mg of the TELN gel precipitate, which is 12% of the total weight.

The results of the SDS-PAGE analysis showed that the proteins in the TELNs were widely distributed, and their protein bands were mainly concentrated in the range of 10–25 kDa (Figure 5a). The protein components of the TELNs were identified using proteomics, and 73 proteins were identified using the UniProt database for the ginger family species. Thirty-five proteins were identical to the protein pool of the Curcuma plant species (Figure 5b). The functions of the proteins were determined using a GO analysis and KOG analysis (Figure 5c,d). The main proteins detected were curcumin synthase 1, curcumin synthase 2, and curcumin synthase 3, which are related to the biosynthesis of secondary metabolites, including curcumin, flavonoids, and terpene skeletons, among others.

### 3.5. Nucleic Acid Determination

The RNAs of the TELNs were mainly distributed in the range of 200–1000 bp (Figure 6a). The PCR identified the RNA species, and the results (Figure 6b) revealed that the TELNs contained essential genes for curcumin biosynthesis that are unique to turmeric plants: curcumin synthetase 1 (CURS1), curcumin synthetase 2 (CURS2), curcumin synthetase 3 (CURS2), and diketo-coenzyme A synthetase (DCS). Meanwhile, the expression of CURS2 within the TELNs was much higher than its expression within the rhizomes of turmeric plants (Figure 6c).

### 3.6. Determination of Chemical Composition

#### 3.6.1. Detection of Chemical Composition by LC-MS

The assay results are shown in Table 3, and the component with the highest relative content within the TELNs was demethoxycurcumin (45.6%), followed by curcumin (7.18%).

#### 3.6.2. Detection of Volatile Components by GC-MS

The results are shown in Table 4. The volatile oil constituent with the highest relative content within the TELNs was turmeric ketone (37.0%), followed by β-gingerone (17.1%).

#### 3.6.3. Quantification of Curcumin by HPLC

The chromatogram of the curcumin detection is shown in Figure 7. Through quantitative analysis, it was found that the TELNs and turmeric rhizomes contained curcumin at concentrations of 30.1 μg/mL and 83.7 μg/mL, respectively, and 0.12 μg of curcumin per 1 mg of TELNs, which accounted for 36.4% of the curcumin content in the rhizomes of its parental plant, turmeric.

## 4. Discussion

In biomedicine, the interest in plant-derived ELNs has significantly increased in recent years, primarily due to their remarkable attributes, such as superior safety, minimal immunogenicity, improved cellular uptake, enhanced gastrointestinal stability, and targeted delivery capabilities. In today’s era of rapid development in nanotechnology, the systematic identification and characterization of ELNs is the only way to promote ELNs to become clinical biotherapeutics and drug carriers.

In this study, the extraction and isolation of ELNs from turmeric rhizome juice was conducted using a traditional approach involving differential centrifugation and ultracentrifugation. This method is commonly employed for the isolation and purification of plant exosomes, along with sucrose density gradient centrifugation, either alone or in combination with ultracentrifugation, and differential centrifugation combined with high-pressure homogenization, to achieve optimal yields and purity [15,16,17]. In this experiment, the turmeric rhizomes were juiced and then centrifuged at a low speed to remove the dead cells and crude fibers. Then, the TELNs were isolated by ultracentrifugation, re-suspended with PBS, and purified by ultracentrifugation again to obtain TELNs with an average particle size of 183.2 ± 10.9 nm. The yield of TELNs can reach about 1.71 ± 0.176 mg/g. However, we tried using kits for exosome extraction and isolation to obtain TELNs, but there were difficulties in scaling up. In a recent study [18], researchers discovered a swift method for isolating plant extracellular vesicles through the utilization of the capillary-channeled polymer fiber spin-down tip technique, and plant-derived extracellular vesicles with an average diameter of 189 nm were obtained from 20 common fruits and vegetables, achieving the low-cost and rapid isolation and extraction of plant exosomes. Nevertheless, there are also some challenges with this approach, notably the utilization of mechanical homogenization, which may compromise the integrity of the cellular structure of the plant material. The existing techniques for isolating and extracting plant exosomes remain relatively limited and intricate, underscoring the methodological innovation’s significance.

Lipids play a significant role in the structural integrity of ELNs’ bilayer membrane and are essential for their stability, uptake, and various biological functions [8]. In the experiment, 615 lipids were identified in the TELNs, of which phosphatidic acid (PA) accounted for 7.22%. Previous research has demonstrated that phosphatidic acid (PA) is highly concentrated in many plant-derived ELNs, leading to cytoskeletal rearrangements and the regulation of proteins involved in vesicular transport/endocytosis, thereby facilitating the uptake of ELNs [19]. Additionally, other lipids with different contents were detected in TELNs, such as phosphatidylethanolamine (PE) and triglyceride (TG). These lipids constitute the rigid bilayer membrane of exosomes and affect the secretion, structure, and signal transduction of exosomes. In addition to PA, TELNs also accumulated PC, with a content of 7.93%. Phosphatidylcholine (PC), present in ginger-derived ELNs (GELNs), has been found to promote the migration of ELNs from the intestine to the liver. More importantly, it has been found that plant ELN lipids are involved in regulating intestinal microbial flora, leading to changes in bacterial composition, localization, and host physiology. For example, PA is an essential lipid for gut *Lactobacillus-rhamnosus*-dependent preferential uptake of GELNs [20]. Therefore, it will be exciting to study the lipids contained in TELNs.

Proteins play a crucial role as essential constituents of biological information in the operation of ELNs. Within plant-derived ELNs, proteins are classified as transmembrane and other plasma-membrane-associated proteins. It was found that the proteins of GELNs were predominantly cytoplasmic proteins, and fewer membrane transport proteins were identified, such as water and chloride channel proteins [21]. This study determined that the proteins contained in TELNs are mainly involved in producing secondary metabolites and are species-specific. For example, curcumin synthase, unique to turmeric plants, can be used as a specific marker for ELNs derived from turmeric, but it lacks universality as a plant exosome marker. Therefore, a large amount of research is currently needed to identify a wide range of plant-derived ELN proteins to reveal their roles in biological and pharmacological activities. RNA is a constituent of biological information derived from plant-derived ELNs, encompassing mRNA, miRNA, and sRNA. Among these, mRNA and noncoding miRNA play a pivotal role in modulating RNA and protein levels within recipient cells, thereby influencing cellular morphology and function [22,23,24]. This study employed the PCR method to ascertain that TELNs possess genes implicated in regulating secondary metabolites, including critical genes for curcumin synthase, aligning with protein species detection findings.

Plant-derived ELNs comprise numerous components, including proteins, lipids, and nucleic acids [13]. These constituents serve biological functions within their original system and play significant roles in organisms through a specific mechanism of cross-border gene expression regulation [13,25,26]. Nanoparticles derived from medicinal and edible plants, such as ginseng, ginger, turmeric, etc., are increasingly being explored for their active effects. According to research, SEO, K. et al. [27] demonstrated that the administration of ginseng-derived extracellular nanovesicles (GDNs) effectively preserved the viability and proliferation of bone-marrow-derived macrophages (BMM) while simultaneously inhibiting the process of osteoclast formation. Additionally, GDNs exhibited significant suppression of the signaling pathways involving IκBα,c-JUN n-terminal kinase, and extracellular signal-regulated kinase, which are induced by RANKL, as well as the genes responsible for regulating osteoclast maturation. Furthermore, plant-derived ELNs exhibit a high concentration of their inherent bioactive compounds. The analysis of GELNs revealed the presence of two distinctive compounds, namely 6-gingerol and 6-gingerol, possessing diverse molecular targets and anti-inflammatory properties. These active components potentially hold therapeutic potential in treating colitis in mice [21]. The curcumin-derived ELNs we studied also detected their active substances, demethoxycurcumin and curcumin. These active substances can exert their anti-inflammatory, antioxidant, and anti-tumor activities to synergistically treat cancer, which is good news for the application of TELNs in the treatment of tumors. The bioactivity and pharmacological functions of plant-derived ELNs are similar to original plants and may be superior to a single active ingredient in plants. This is attributed to the synergistic effect of the nanoscale size and multiple components in plant-derived ELNs, which makes plant-derived ELNs have higher oral bioavailability and barrier permeability [28]. Therefore, the potential biological and pharmacological activities of TELNs, derived from plants, are highly anticipated. The presence of native chemically active substances in plant-derived ELNs confers a distinct advantage, distinguishing them from exosomes derived from alternative sources and broadening the scope of their potential applications.

In recent years, there has been a notable shift in the field of nanotechnology towards environmentally friendly and sustainable approaches. This transformation is also evident in nanomedicine and drug delivery [29]. Within the medical field, for instance, using SC-EVs in conjunction with bioactive materials such as hydrogels or scaffolds for bone tissue engineering purposes has proven to be highly efficacious in stimulating osteogenesis and mending bone imperfections [30]. In drug delivery systems, nanoparticles are widely used as a new generation of drug carriers. However, research on their immunogenicity, cytotoxicity, and other aspects has always been the focus of attention. Karami, Z. et al. [31] developed and characterized the monocyte-based NG platform in which a biomimetic monocyte-derived cell membrane surrounded the core GL-loaded PLGA nanoparticle. Exosomes, a naturally occurring nanoscale vesicular structure possessing favorable biocompatibility and safety attributes, have garnered attention in drug delivery due to their potential as efficient carrier materials capable of traversing biological barriers [32]. In this study, TELNs derived from turmeric as a medicinal and edible source possess low immunogenicity and can achieve immune evasion. Moreover, TELNs originate from simple sources and are cost-effective. Exosomes exhibit remarkable deformability and exceptional permeability, rendering them suitable for transporting diverse small-molecule drugs and macromolecules, enhancing the transdermal and dermal retention of medications and augmenting their local efficacy and adherence to drug administration [33]. Consequently, there has been a growing interest in investigating the potential of plant-derived ELNs as drug delivery systems. Gao, C. et al. [12] isolated and purified exosomes from turmeric juice, developed turmeric-derived nanovesicles (TNV) to alleviate colitis, and explored its potential mechanism of action. Yang, X. et al. [34] used turmeric-derived exosome-like nanoparticles to load astragalus components, which significantly promoted the uptake and transport of astragalus and produced better antitumor effects than using astragalus alone. Although this study only conducted a component analysis and corresponding characterization of TELNs, as plant-derived exosome-like nanoparticles that are homologous to medicine and food, the comprehensive analysis of TELNs not only stripped back the “mysterious veil” of their components but also lays a solid foundation for their application as natural drug carriers with low immunogenicity and low toxicity.

## 5. Conclusions

In this study, we successfully extracted and isolated ELNs from turmeric of the medicinal plant turmeric using ultracentrifugation at an extraction rate of 1.71 ± 0.176 mg/g. A systematic and comprehensive physicochemical analysis of the TELNs revealed that they possessed the key genes for curcuminoid synthesis, such as curcumin synthase (CURS), contained phosphatidylethanolamine (PE) (17.4%), triglyceride (TG) (12.3%), and other lipid-rich components, as well as proteins involved in secondary metabolite biosynthesis. This study represents the inaugural combination of multi-omics to analyze the characteristics and nature of the TELNs, which furnishes a significant scientific foundation for the subsequent advancement and exploitation of turmeric.

## Figures and Tables

**Figure 1 bioengineering-10-01199-f001:**
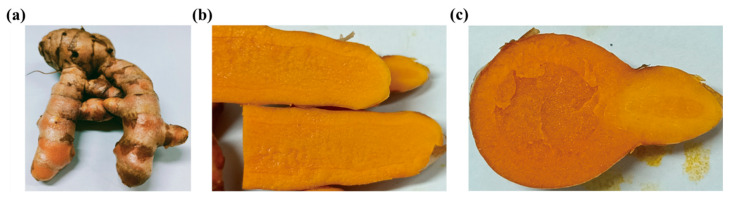
Morphological identification of turmeric. (**a**) Morphological appearance; (**b**) transverse section; (**c**) longitudinal section.

**Figure 2 bioengineering-10-01199-f002:**
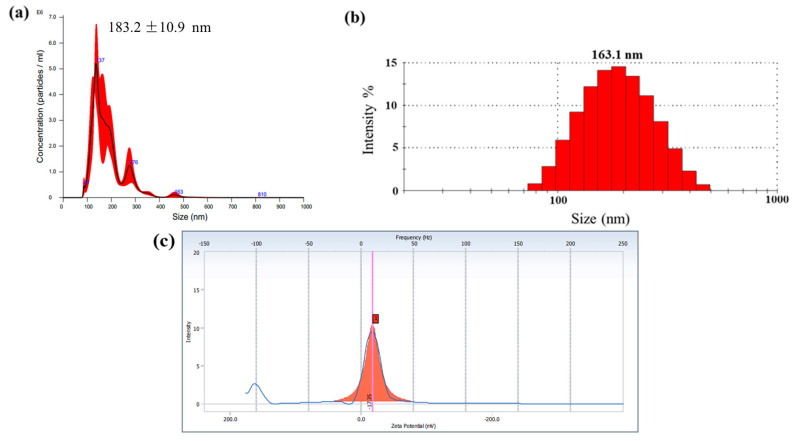
Particle size distribution and zeta potential of the TELNs (*n* = 3). (**a**) Particle size distribution diagram of the NTA detection of TELNs; (**b**) particle size distribution diagram of the DLS detection; (**c**) zeta potential of the TELNs.

**Figure 3 bioengineering-10-01199-f003:**
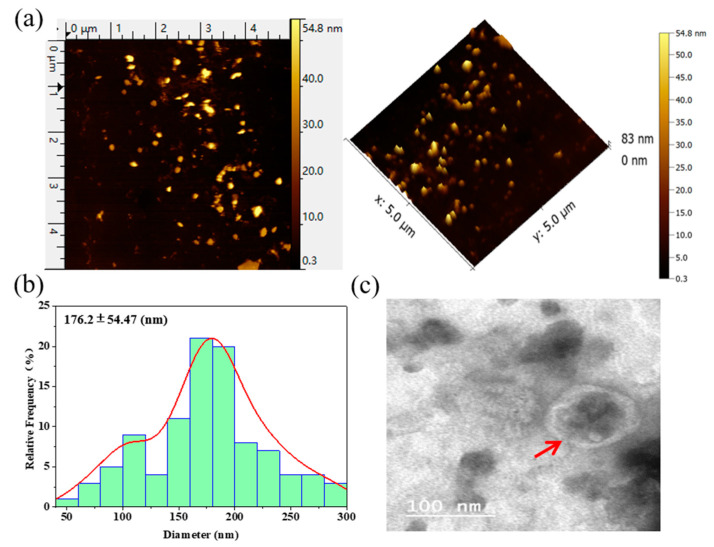
The appearance, morphology, and structure of TELNs. (**a**) AFM image of the appearance and morphology of TELNs; (**b**) columnar curve fitting of TELNs size; (**c**) TEM structural diagram of the TELNs.

**Figure 4 bioengineering-10-01199-f004:**
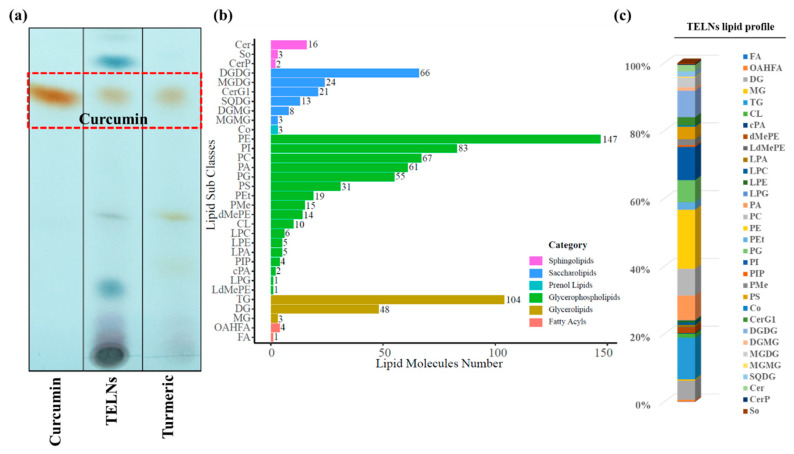
Analysis of lipid composition of TELNs. (**a**) TLC analysis of total lipids in TELNs and turmeric plants; (**b**) analysis of total lipids in TELNs using lipidomics; (**c**) statistical analysis of lipid component content.

**Figure 5 bioengineering-10-01199-f005:**
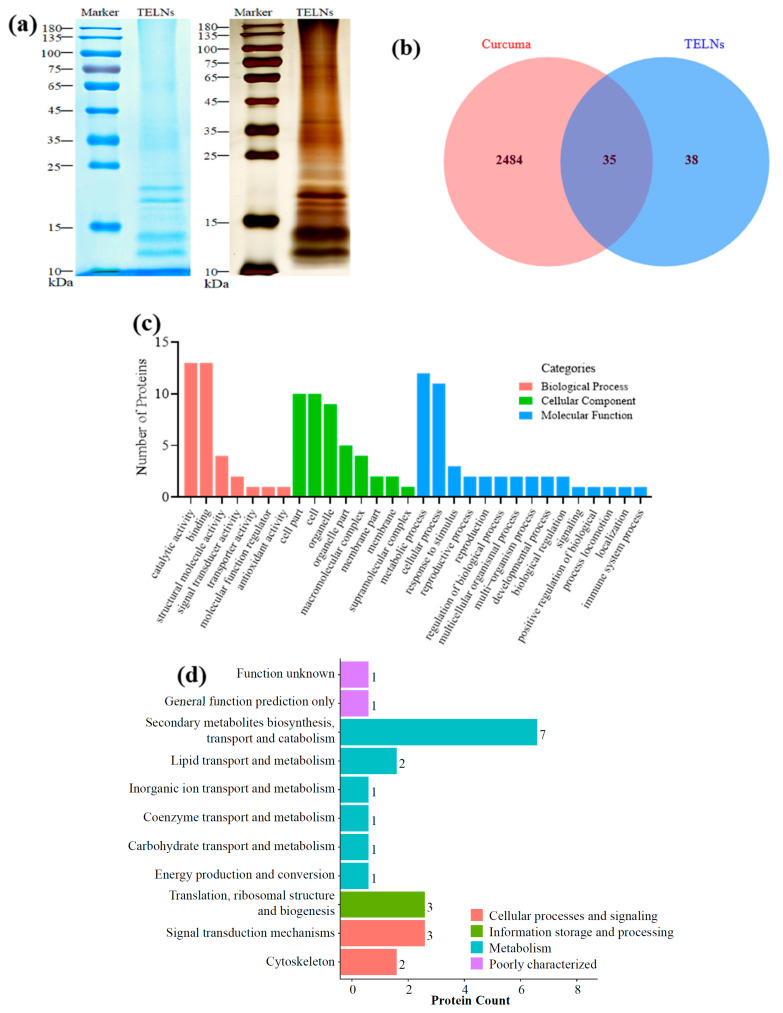
Analysis of the protein composition and function of TELNs. (**a**) Total protein analysis of TELNs using SDS-PAGE; (**b**) identification of total proteins in TELNs and turmeric using proteomics; (**c**) GO function analysis of TELN proteins; (**d**) KOG function analysis of TELN proteins.

**Figure 6 bioengineering-10-01199-f006:**
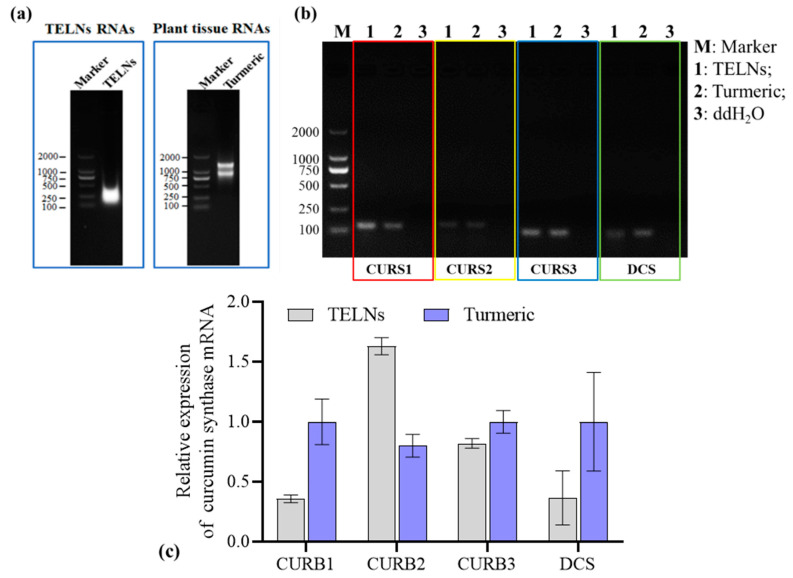
Analysis of the nucleic acid information of TELNs. (**a**) Agarose gel electrophoresis results from analyzing the total RNA of TELNs and turmeric; (**b**) PCR results from identifying RNA species in TELNs; (**c**) RT-qPCR results from detecting the expression of key genes in curcumin biosynthesis (*n* = 3).

**Figure 7 bioengineering-10-01199-f007:**
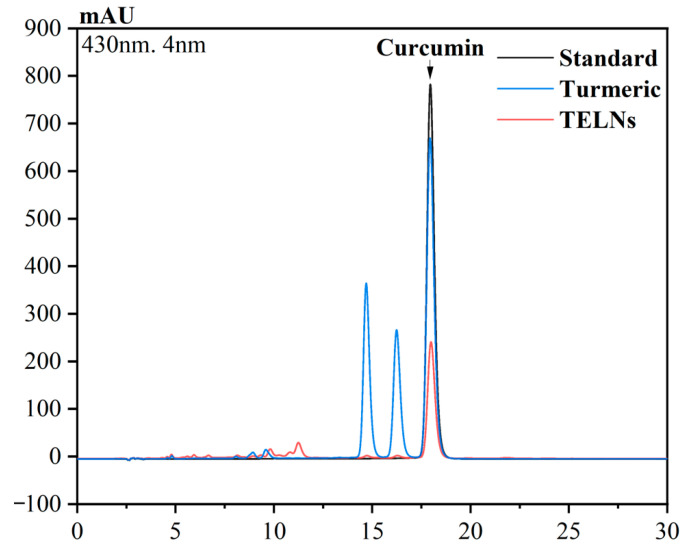
Quantification of curcumin using HPLC.

**Table 1 bioengineering-10-01199-t001:** Primer information.

Gene Name	Primer Sequence (5′–3′)	Size (bp)
DCS	F: GTGCTGTTCATCCTGGACGAG	21
	R: CAACAGCACGCCCCAGTCGA	20
CURS1	F: CATCATTGACGCCATCGAAGC	21
	R: TCAGCTCATCCATCACGAAGTACAC	25
CURS2	F: TAGGGATCAAGGACTGGAACAAC	23
	R: TGTTGCCGAACTCGGAGAAGAC	22
CURS3	F: TGGAGCCCTCCTTCGACGACC	21
	R: CCCATTCCTTGATCGCCTTTTCC	23
Actin	F: GGATATGCTCTTCCTCATGCT	21
	R: TCTGCTGTGGTGGTGAATGA	20

**Table 2 bioengineering-10-01199-t002:** The amount of protein in the TELN gel precipitate (*n* = 7).

Groups	1	2	3	4	5	6	7	Average Protein Amount (mg/mg)
TELN gel precipitate amount (mg)	56.0	58.0	63.0	66.0	67.5	57.4	68.0	0.125 ± 0.0162
TELN protein amount (mg)	6.50	8.75	7.81	7.32	7.25	7.04	9.69	

**Table 3 bioengineering-10-01199-t003:** Analysis of the chemical composition of TELNs.

Number	Retention Time (min)	Chemical Substance	Concentration (%)
1	5.57	Bisacurone	1.72
2	7.66	Curcumin	2.47
3	8.44	α-Curcumene	1.02
4	8.84	Trans-Anethole	0.32
5	8.99	Turmerone	2.07
6	9.6	Bisabolol	1.25
7	10.08	Curcumenol	7.18
8	10.2	2,5-Dihydroxybisabola-3,10-diene	0.80
9	11.59	Sitosterol	2.53
10	11.81	Demethoxycurcumin	45.63
11	14.08	Zedoarondiol	1.51

**Table 4 bioengineering-10-01199-t004:** Analysis of volatile oil components of TELNs.

Number	Retention Time (min)	Chemical Substance	Concentration (%)
1	7.97	α-Phellandrene	3.01
2	8.58	O-Cymene	0.84
3	10.5	Cyclohexene	0.66
4	24.9	α-Curcumene	1.53
5	25.5	β-Curcumene	5.42
6	26.8	β-Sesquiphellandrene	2.98
7	33.8	Ar-Turmerone	13.7
8	34	Turmerone	37
9	35.5	β-Turmerone	17.1
10	94.4	Sitosterol	0.46
11	96.1	γ-Sitosterol	2.32

## Data Availability

Not applicable.

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
