# Peer review of "Extraction, Isolation, and Component Analysis of Turmeric-Derived Exosome-like Nanoparticles"

_bioengineering, 2023, doi:10.3390/bioengineering10101199_

Round 1
Reviewer 1 Report
Dear editor
In this study, authors have tried to extract the extracellular lipid nanoparticles (ELNs) from turmeric rhizome juice using centrifugation and ultracentrifugation and characterized the exosome like carrier in terms of peptides/proteins, lipids and chemical compositions as well as curcumin content. The steps in design and presenting data are logical but it is poorly presented regarding English grammar. Importantly, author have to explain the novelty of their work compared to published articles including doi: 10.7150/thno.73650 and https://doi.org/10.1016/j.jddst.2023.104274
.However, authors should address these questions:
1. Why do authors have not characterized the exosome-like nano-particles (TELNs) in term of carbohydrate.
2. Authors must report size dispersity as PDI in the results.
3. Are authors sure that there was not phosphatidyl choline in the composition of the TELNs? If there is, why they have not mentioned it as a main PC.
4. The method of TELNs extraction should be mentioned in the abstract.
5. I last sentence of the abstract, authors should explain the main benefits or application of the study for example: a promising drug carriers with reduced immunogenicity and toxicity.
6. Is curcumin-like compounds mean curcumin derivatives chemically? Authors should mention the advantages of the exosome-like nanoparticles compared to other membtane based nanoparticles like nanoghost. Authors can cite some articles like: https://doi.org/10.1038/s41598-023-41136-y
7. There are some grammatical mistakes in the text must be corrected like space between …..[3].F…. in line 47.
8. No need to report the results of the study in details in lines 81-95. Usually, the procedure must be mentioned in general. Please correct and revise it.
9. The sentence has been repeated twice in lines 123-124: The gel precipitate was added to PBS and 123 resuspended by sonication. The gel precipitate was added to PBS and resuspended by 124 ultrasonication, .. Correct it.
The sentence in lines 121-125 don’t have continuity, please revise them to make them understandable.
10. The manuscript must be fully edited in terms of grammer and adjustment.
11. In TELNs preparation procedure, authors have not mentioned the lyophilization step.
12. Revise the lines of 153-176 in a scientific manner (refer to authors guideline) as well as 2.5.2 and 2.5.3 sections.
13. Statistical analysis must be added to the materials and methods.
14. Fig. 2a,c should be replaced by original data, taken from instruments.
15. There is no index for size dispersity.
16. How can authors show a TEM image having numbers of TELNs?
17. The resolution of Fig.5 must be improved.
18. In Figure 6, the negative error bar is not obvious.
19. Authors should discuss the reasons for preference abundance of the certain phospholipids.
20. Authors must be have a literature survey to compare the TELNs with others. They should mention the limitation and future respective.
Best
Abstract should be enriched with more results.
2. Authors should describe the significance of personalized medicine and tailored liver models in introduction.
3. Describe “*” below fig.1 in its caption.
4. For the mentioned purpose, 3D printing technique is so critical. How is it possible to use 3D model of liver with dedicated anatomy by some 3D printing techniques like FBM? Some 3D printing technique should be put into exclusion criteria especially for surgeon guidance, as the 3DPLMs is more appropriate for highly complex or laparoscopic case.
Additionally, authors have to distinguish the application of the 3D models for patient education and surgeon navigation.
Best
Extensive editing of English language required which have been mentioned in the comments.
Author Response
请参阅附件。

Reviewer 2 Report
The characterization of exosome is great, but the novelty is not completely mentioned in the manuscript, and the writing errors are not less, please note that. In addition, here are some other suggestions that the authors should consider.
Major
1. The novelty is not mentioned either in the abstract or conclusion parts. The authors should use the words like “we first discovered/detected/extracted……“.
2. In the last paragraph of the Introduction part, it’s repeated with the abstract, I suggest re-writing this part with lots of revisions.
3. Why is the size in the AFM image larger than DLS? Usually, the hydrated particle size should be larger than the real size. And how did you analyze the size of the AMF image, please list the process in the experimental part.
Minor
1. Line 16, It’s better to change “resolved” to “detected”.
2. Line 23, Does the synthesis of curcumin need a gene? Please offer the reference.
3. Line 27, 0.1% should be 0.17%.
4. In vivo and in vitro should be italicized.
5. Please supplement the version and model of instruments in the 2.1 part.
6. I’d suggest the authors draw the figure by themselves, not using the original figure (like Fig. 1(b) (c), and Fig. 7) to make the figures clear and beautiful.
7. Line 268, blank between word and brackets.
8. Line 284, the full name of TLC should be listed.
9. Line 351 and 360, delete “(ELNs)”.
Lots of grammar errors, please check really carefully.
Round 2
Reviewer 1 Report
.
.